# Experimental Testing of Bandstop Wave Filter to Mitigate Wave Reflections in Bilateral Teleoperation

**Isaac O. Ogunrinde** [1], **Collins F. Adetu** [2], **Carl A. Moore Jr.** [3,*], **Rodney G. Roberts** [1] and **Keimargeo McQueen** [4]

1  Department of Electrical and Computer Engineering, FAMU-FSU College of Engineering, Tallahassee, FL 32310, USA; isaac1.ogunrinde@famu.edu (I.O.O.); rroberts@eng.famu.fsu.edu (R.G.R.)
2  Ronler Acres Campus, Intel Corporation, Hillsboro, OR 97124, USA; collins1.adetu@famu.edu
3  Department of Mechanical Engineering, FAMU-FSU College of Engineering, Tallahassee, FL 32310, USA
4  Department of Computer Science, Florida Agricultural & Mechanical University, Tallahassee, FL 32307, USA; keimargeo1.mcqueen@famu.edu
*  Correspondence: camoore@eng.famu.fsu.edu; Tel.: +1-(850)-410-6367

**Abstract:** A bilateral teleoperation system can become unstable in the presence of a modest time delay. However, the wave variable algorithm provides stable operation for any fixed time delay using passivity arguments. Unfortunately, the wave variable method produces wave reflection that can degrade teleoperation performance when a mismatched impedance exists between the master and slave robot. In this work, we develop a novel bandstop wave filter and experimentally verify that the technique can mitigate the effects of wave reflections in bilaterally teleoperated systems. We apply the bandstop wave filter in the wave domain and filtered the wave signal along the communication channel. We placed the bandstop wave filter in the master-to-slave robot path to alleviate lower frequency components of the reflected signal. With the lower frequency components reduced, wave reflections that degrade teleoperation performance were mitigated and we obtained a better transient response from the system. Results from our experiment show that the bandstop wave filter performed better by 67% when compared to the shaping wave filter respectively.

**Keywords:** bilateral teleoperation; filtering; time delay; passivity; wave variable; wave reflections

## 1. Introduction

Teleoperation involves controlling, operating, and manipulating remote robots and/or systems, in hostile environments, or in helping humans to accomplish strenuous tasks [1–4]. Since its conception, teleoperation has been applied in various areas such as space exploration [5], surveillance [6,7], volcano exploration, landmine detection [8], search and rescue [9], robotic surgery [10–12], mobile robots [13–16], and dealing with corrosive and deadly materials or substances [17]. The three main components of a teleoperation system are the master device, the remote or slave device, and the communication channel. Unilateral or bilateral teleoperation are two classifications of remote teleoperation. In unilateral teleoperation, the operator is not provided with force feedback from the slave device; rather, the operator uses visual and auditory cues of the slave device condition to supervise its interaction with the environment. In contrast, for bilateral teleoperation, the master device receives force feedback from the slave device. Force feedback enables the operator to "feel" the remote environment which enables improved manipulation and better monitoring of the interaction between the slave device and the environment [18]. However, the presence of time delay in the communication channel connecting the master and remote robots threatens the quality of the force feedback provided in the bilateral teleoperation system. It was demonstrated in previous studies that the introduction of the

modest time delay could cause the instability of a bilateral teleoperation system [19–22]. An example of a real-life application where the presence of time delay can be problematic is a teleoperated combat drone. The instability induced due to time delay of a fraction of a second in shooting a missile could cause that missile to hit the wrong target.

Several solutions to time delay induced instability in bilateral teleoperation were proposed in the literature. Leung et al. [23], suggested a tunable H∞ optimal-based controller by which a teleoperation system can gain stability for a prescribed time delay margin, with more improvement presented by Sano et al. [24]. Extending the work proposed by Anderson and Spong, Neimeyer and Slotine developed the wave variable method, which modeled the bilateral teleoperation system as an ideal power transmission line with a more mechanical formulation [25]. The wave variable method assures the stability of the bilateral teleoperation system in the presence of any fixed time delay using passivity arguments. Nonetheless, the fixed time delay problem is still an important problem and recent research shows that [26–28]. Shukla et al. [26] designed a passive sliding mode controller that exhibits Real Time Cyber-Physical System characteristics for trajectory tracking of a mobile robot in the presence of fixed time delay and traffic disturbances.

However, the wave variable method exhibits several intrinsic drawbacks which include position drift, wave reflections, etc. Our focus in this paper is wave reflections. The wave variable method produces wave reflections when impedance mismatch occurs between the master and slave site, which can cause system performance to deteriorate [29–31]. Sun et al. in [32] conducted a comprehensive survey on wave variable controlled bilateral teleoperation systems. The authors claimed that wave reflections can be exposed by evaluating the standard wave-based teleoperation architecture. A wave controller-based teleoperation system can have three independent signal feedback channels. Channel 1 is made up of the feedback from the master in the form of the damping. In channel 2, the operator receives the feedback from the slave and channel 3 consists of the wave reflections. Wave reflections in channel 3 can degrade the information present in channel 1 and 2 if it is not removed. The continual reflection of the wave signal develops an oscillatory behavior of the bilateral teleoperation system which is a distraction to the system operator [33]. We can compare these wave reflections to echoes generated in a deficient acoustic system. Tian et al. in [34], evaluated wave reflection in the frequency domain and concluded that wave reflection occurs as a result of oscillation in the response of the teleoperation system.

Various approaches were suggested in the literature to attenuate the wave reflections that degrade teleoperation performance [35–39]. Bate et al. in [40] developed a wave-based method that can reduce wave reflections and maintain stability in teleoperation for unknown environments. This was achieved by altering the wave variables architecture without necessarily satisfying the impedance matching requirement while stability is guaranteed. Sun et al. in [41] developed a four-channel (4-CH) architecture that made use of two modified wave transformation schemes for improving wave-based system transparency and assuring stability under time-varying delays. The two modified wave transformation schemes were implemented such that the feed-forward signals were encoded with the feedback signals. To address the issue of uncertainties in the system, a sliding mode control algorithm was implemented. With the modified 4-CH architecture, wave reflections were mitigated, the signal transmitted was improved while satisfying passivity argument. Zeng et al. in [42] suggested a wave-based bilateral teleoperation approach using time domain passivity control to mitigate wave reflection and improve tracking performance. However, the two major categories for attenuating wave reflections are impedance matching and filtering techniques. Impedance matching is a technique in which the wave impedance of the master site is set to match the impedance of the slave site. This can be achieved when additional damping is introduced into the system [30,31,43]. It is known that this approach attenuates wave reflections, yet it slows down the feedback coming from the remote site to the operator [38]. Likewise, there can also be a difficulty in directly matching impedances because the knowledge of the impedance of the remote environment is required [37]. Sometimes the remote site impedance changes continuously and/or it may be that the remote environment is inaccessible.

Filtering is another technique used in mitigating wave reflections. Previous works considered the time delay across the communication channel to be mainly associated with the frequency at which these wave reflections occur. First, Niemeyer and Slotine in [44] suggested the use of low pass wave filters to attenuate wave reflections. Selecting the cutoff frequency of the low pass filter depends on the time delay across the communication channel. The drawback of implementing a low pass filter is that it gravely limits the system bandwidth, although it does decrease the wave reflection. Hirche and Buss in [45] proposed developing filters in the frequency domain by implementing an optimization algorithm. The use of the algorithm requires prior knowledge of the system's impedance, so it becomes more difficult when dealing with varying impedances. Kuschel in [46] continued this work and introduced the use of a bank of filters with an impedance matching ability of several systems, using smoothing functions to alternate between different filters. Ye et al. in [47] developed a wave variable-based approach that enhances the fidelity of haptic feedback and eliminates force reflection bias. To enhance the fidelity of haptic feedback, the passivity of the system diminishes. To guarantee the passivity of the teleoperated system, the authors implemented a low pass filter with a bandwidth that can be tuned.

Furthermore, Tanner et al. in [38] proposed that only the resonant frequency should be targeted rather than implementing a low pass filter that limits the system's bandwidth manipulation. The authors presented the infinite and finite shaping wave filters that depend on the time delay that exists in the communication channel. However, the shaping wave filters are inadequate when the frequency of wave reflection drift based on the changes to system dynamics of the teleoperated system. In our study, we discovered that changes in the dynamics (slave mass) of a teleoperated system can affect the frequency components of wave reflections. These frequency components consist of lower and higher orders that can drift based on the changes to system dynamics of the teleoperated system. Besides, we observed that the increase in slave mass increases the magnitude of the reflected signal of wave reflections. Despite the success of the shaping wave filters with respect to time delay, the method struggles when the size of the slave mass is increased. This is because the shaping wave filters mitigate wave reflections based on the time delay only without considering changes to system dynamics. Nonetheless, to our knowledge, no previous research has examined this problem.

In this work, we develop a wave-variable-based controller (bandstop wave filter). In addition, we experimentally verify that the technique can mitigate wave reflections in the presence of fixed time delay and changes to system dynamics of the teleoperated system.

We apply the bandstop wave filter in the wave domain and filtered the wave signal along the communication channel. We placed the bandstop wave filter in the master-to-slave robot path to alleviate lower frequency components of the reflected signal. With the lower frequency components reduced, wave reflections that degrade teleoperation performance were mitigated and we obtained a better transient response from the system. Results from our experiment show that the bandstop wave filter performed better by 81% and 67% when compared to the traditional wave variable method and shaping wave filter respectively.

The rest of this paper is structured as follows. In Section 2, we introduce the wave variable algorithm and the effect of wave reflection on teleoperation performance. In Section 3, we discuss wave reflections and how their frequency characteristics can be affected by changing the slave mass. In addition, we present shaping wave filters one of the solutions proposed in the literature to address wave reflection. In Section 4, we present the bandstop wave filters and its ability to attenuate wave reflections. In Section 5, we illustrate the experimental setup and present our analyzed results. Finally, Section 6 covers the conclusion.

## 2. Wave Variable Algorithm and Passivity

A bilateral teleoperation system with time delay across its communication channel gains stability when the wave variable algorithm is applied. In this section, we show how the wave variable

provides stability to a bilaterally teleoperated system with time delay using passivity. In addition, we demonstrate the lingering effects of wave reflection on bilaterally teleoperated systems.

The wave variable method is implemented such that the power variables $F$ and $\dot{x}$ are transformed into the wave variables $u$ and $v$ before being transmitted across the communication channel. We present the transformations as follows:

$$u_m(t) = \frac{1}{\sqrt{2b}}\left[b\dot{x}_m(t) + F_m(t)\right] \tag{1}$$

$$v_m(t) = \frac{1}{\sqrt{2b}}\left[b\dot{x}_m(t) - F_m(t)\right] \tag{2}$$

$$u_s(t) = \frac{1}{\sqrt{2b}}\left[b\dot{x}_s(t) + F_s(t)\right] \tag{3}$$

$$v_s(t) = \frac{1}{\sqrt{2b}}\left[b\dot{x}_s(t) - F_s(t)\right] \tag{4}$$

where $m$ represents master and $s$ represent the slave, and b denotes the characteristic wave impedance which for one degree-of-freedom is a positive constant, but it is a symmetric positive definite matrix for multiple degree-of-freedom [48]. The transmitted wave variables are:

$$u_s(t) = u_m(t - T), \; v_m(t) = v_s(t - T) \tag{5}$$

Figure 1 illustrates wave transformation following its implementation across the communication channel of a bilateral teleoperation system consisting of constant time delay $T$.

On arriving at the slave side of the teleoperation system, the wave variables are converted back to the following power variables:

$$\dot{x}_m(t) = \frac{1}{\sqrt{2b}}\left[u_m(t) + v_m(t)\right] \tag{6}$$

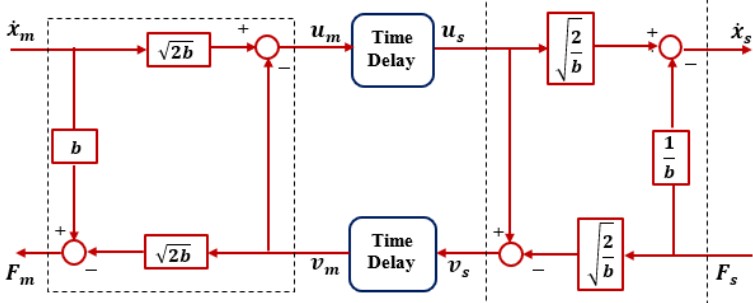

**Figure 1.** A communication channel with wave variables across it.

$$F_m(t) = \sqrt{\frac{b}{2}}\left[u_m(t) - v_m(t)\right] \tag{7}$$

$$\dot{x}_s(t) = \frac{1}{\sqrt{2b}}\left[u_s(t) + v_s(t)\right] \tag{8}$$

$$F_s(t) = \sqrt{\frac{b}{2}}\left[u_s(t) - v_s(t)\right] \tag{9}$$

The flow of power, $P_{in}(t)$ into the communication channel can be expressed as [25]:

$$P_{in}(t) = \dot{x}_m^T(t)F_m(t) - \dot{x}_s^T(t)F_s(t) \tag{10}$$

Substituting (6), (7), (8) (9) into (10), the flow of power $P_{in}(t)$ is expressed as follows [25,35]:

$$P_{in}(t) = \frac{1}{2}\Big(u_m^T(t)u_m(t) - v_m^T(t)v_m(t) - u_s^T(t)u_s(t) + v_s^T v_s(t)\Big) \tag{11}$$

Integrating the result obtained when Equation (5) is substituted into (11) and assuming $E_{store}(0)$ is zero, the energy stored when wave variable is implemented can be expressed as follows:

$$\int_0^t P_{in}(\tau)d\tau = E_{store}(t) = \frac{1}{2}\int_{t-T}^t \Big[u_m^T(\tau)u_m(\tau) + v_s^T(\tau)v_s(\tau)\Big]d\tau \geq 0 \tag{12}$$

Hence, the stored energy is perpetually non-negative which guarantees the passivity of the communication channel when the wave variable algorithm is implemented.

Illustrated in Figure 2a is the position, velocity, and force of a simulated bilateral teleoperation system whose master velocity was set to a step input of 0.05 m/s with a time delay of 100 ms. It is evident that the system is unstable because the position, velocity and force curve are unbounded. The system shown in Figure 2b is the same as Figure 2a but has the wave variable method implemented. In contrast to the unbounded behavior of the system in Figure 2a, the system in Figure 2b has a bounded behavior which implies that the system is stable. Although the wave variable algorithm could provide stability for teleoperation systems with time delay. Nonetheless, the algorithm produces transient oscillations, called wave reflections, which can degrade teleoperation performance as shown in Figure 2b. Wave reflection can be generated in a bilateral teleoperation system when the incoming wave signals are being reflected. Another shortcoming of the wave variable algorithm is position drift error. Position drift caused by steady-state position errors due to integration because the encoded parameter in the communication channel is velocity and not position. However, in this paper, we focus on wave reflections.

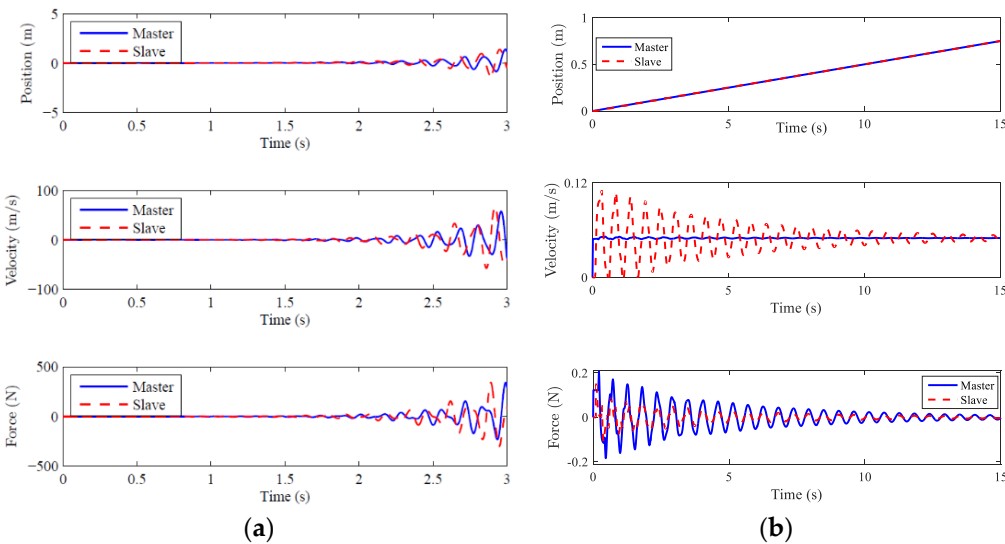

**Figure 2.** (**a**) Plot showing the response of the master and slave robots with 100 ms delay before using wave variable (**b**) wave reflections with wave variable implemented at a time delay of 100 ms.

## 3. Wave Reflections and Shaping Wave Filter

In Section 2, we stated that the wave variable algorithm, when applied in a bilateral teleoperation system, produces wave reflections caused by incoming wave signals that were reflected. In this section, we discuss the cause of wave reflections and how the frequency characteristics of wave reflections can be affected by changes in the dynamics of a teleoperated system.

### 3.1. Wave Reflections

In Section 1, we mentioned several intrinsic drawbacks of wave variable algorithm which includes wave reflections and we discussed wave reflections in detail. However, wave reflections occur when impedance mismatch exists between the master site and the slave site, such that the incoming wave signals can only be partially absorbed by the master or slave site [29–31]. This can cause the performance of the teleoperated system to deteriorate [32].

Wave signals can travel within the bilateral teleoperation system using several paths as shown in Figure 3. Studies showed that these paths are peculiar to the wave variables. Velocity $\dot{x}_m$, and the wave impedance $b$ is the apparent damping term which develops wave signal, $u_m$ described in Equation (13). Feedback signals from the slave travel back to the operator using the slave feedback path shown in Figure 3. It is most preferable that no other signal should corrupt the slave feedback signal as it provides the operator with a credible feedback force from the slave device. However, this is not true for wave variable algorithm as wave reflections travel through the master and slave device communication channel. The outgoing wave signal $u_m$ on the master robot site consist of the incident signal $v_m$ that can be reflected if not absorbed completely.

$$u_m(t) = \dot{x}_m(t)\,\sqrt{2b} - v_m(t) \tag{13}$$

Similarly, outgoing wave $v_s$ described in Equation (14) on the slave site consist of the incident wave signal $u_s$ such that if not absolutely absorbed by the slave device, some fraction of it will be reflected. If the wave reflections linger on the communication channel they can result in transient oscillations and inefficient feedback force.

$$v_s(t) = u_s(t) - \sqrt{\frac{2}{b}}F_s(t) \tag{14}$$

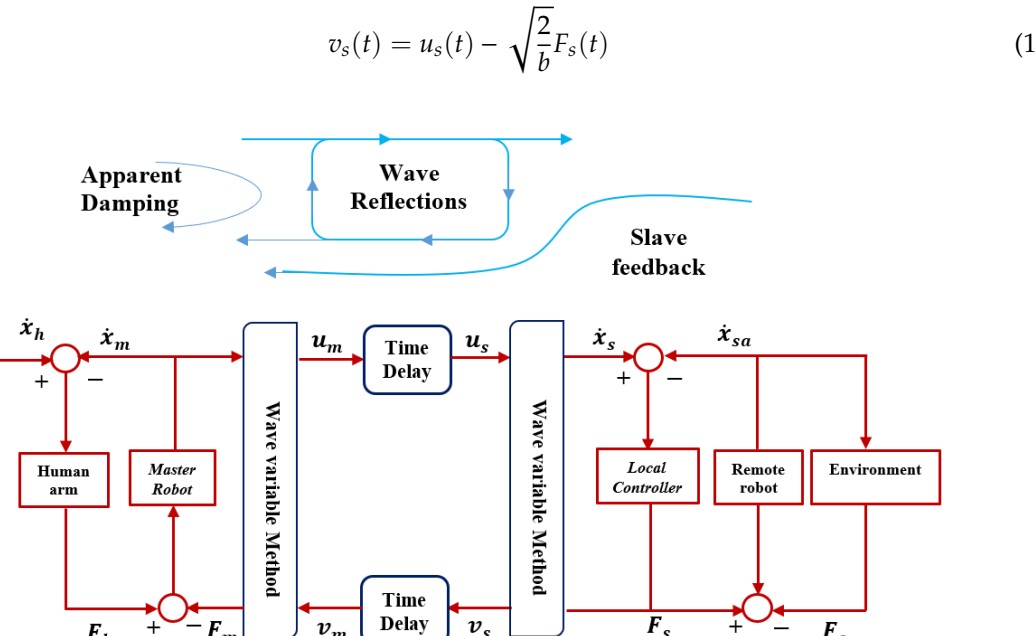

**Figure 3.** Paths traveled by wave signal when the wave variable is employed in a bilaterally teleoperated system.

### 3.2. Shaping Wave Filter

In Section 1, we mentioned that impedance matching and filtering techniques are the two major approaches suggested in the literature to attenuate wave reflections. Our focus is the filtering technique. In previous works, several authors considered the time delay to be mainly associated with frequency at which wave reflections cause system performance to deteriorate [31,35,38,45].

Tanner et al. in [38] proposed that only the resonant frequency should be targeted rather than implementing a low pass filter that limits the system's bandwidth manipulation. The authors presented the infinite and finite shaping wave filters that absolutely depend on the time delay that exists in the communication channel. The transfer functions of the finite and infinite filters are outlined as:

$$G_{\text{finite}}(s) = \frac{1 + e^{-2sT}}{2} \tag{15}$$

$$G_{\text{infinite}}(s) = \frac{1}{2 - e^{-2sT}} \tag{16}$$

The presence of $e^{-2sT}$ in Equations (15) and (16) enables the finite and infinite filters to mitigate the dominant frequency components of the wave reflections for limited cases, and it only depends on the time delay across the communication link. Finite and infinite filters are unsuitable when the frequency of wave reflection drift based on the changes to system dynamics of the teleoperated system [36]. However, the shaping wave filters are inadequate when the frequency of wave reflection drift based on the changes to system dynamics of the teleoperated system. In our study, we discovered that changes in the dynamics (slave mass) of a teleoperated system can affect the frequency components of wave reflections. These frequency components consist of lower and higher orders that can drift based on the changes to system dynamics of the teleoperated system. In addition, we observed that the increase in slave mass increases the magnitude of the reflected signal of wave reflections.

### 3.3. Simulation Results

Table 1 contain teleoperation parameters used in our simulations [35,38,49]. In this section, we compared simulation results between traditional wave variables and shaping wave filter.

**Table 1.** The parameter used for simulating the bilateral teleoperation system.

| Parameter | Value |
|---|---|
| Human arm spring constant, $K_h$ | 75 N/m |
| Human arm damper constant, $B_h$ | 50 Ns/m |
| Slave controller proportional gain, Kc | 370 N/m |
| Slave controller derivative gain, Bc | 2.5 Ns/m |
| Wave impedance, b | 2.5 Ns/m |
| Time delay, T | 100 ms |

Using Figure 3, we could illustrate the effect of change in dynamics (slave mass) on wave reflections. The one degree-of-freedom mathematical model for the master and slave robot system used in our simulations is as shown below:

$$M_m \ddot{x}_m = F_h(t) - F_m(t) \tag{17}$$

$$M_s \ddot{x}_s = F_s(t) - F_e(t) \tag{18}$$

where $M_m$ denotes the master device mass, $F_h(t)$ represents the human force, $F_m(t)$ is the force applied to the master device, $\ddot{x}_m$ is the acceleration of the master device, $M_s$ denotes the mass of the slave device, $F_s(t)$ represents the force due to the PD controller applied to the slave device, $F_e(t)$ is the force applied on the slave's end effector when in interaction with the environment, $\ddot{x}_s$ is the slave device acceleration.

The computation of $F_s(t)$ the force due to the PD controller applied to the slave device used in our simulations:

$$F_s(t) = K_c(x_s - x_{sa}) + B_c\left(\dot{x}_s - \dot{x}_{sa}\right) \tag{19}$$

where $K_c$ denotes the proportional gain and $B_c$ denotes the damping gains, $s$ is the commanded slave and $sa$ is the actual slave signal.

For our simulations, we adopted the human force that has a spring-damper model of the human arm modeled as a proportional derivative (PD) position tracking controller from [36,49,50]:

$$F_h(t) = K_h(x_h - x_m) + B_h\left(\dot{x}_h - \dot{x}_m\right) \tag{20}$$

where the human intended position and velocity are denoted by $x_h$ and $\dot{x}_h$ respectively.

In the literature, the damping constant $B_h$, and spring constant $K_h$ used for simulation were set to 75 N/m and 50 N/m respectively [36,49,50]. Implementing the damping and spring constants of the human arm model provided an over-damped response which makes the system feel like the human operator positions the master robot without overshoot [50]. We set the command velocity of the master device to 0.05 m/s Figure 4 shows the fast Fourier transform (FFT) of the feedback force after applying the shaping wave filter (Figure 4b) and we compare the result in that of the traditional wave variable method (Figure 4a).

Slave masses 0.1 kg and 0.5 kg were used for the simulation with a time delay of 100 ms. From Figure 4a, the highest magnitude of the reflected signal is 0.8 when the wave variable algorithm was implemented. Using the shaping wave filter the dominant frequencies of the wave reflection were mitigated for both the simulated slave masses. The highest reflected magnitude when shaping wave filter was employed is 0.45 as shown in Figure 4b. The magnitude of wave reflection decreased by 44% when the shaping wave filter was implemented.

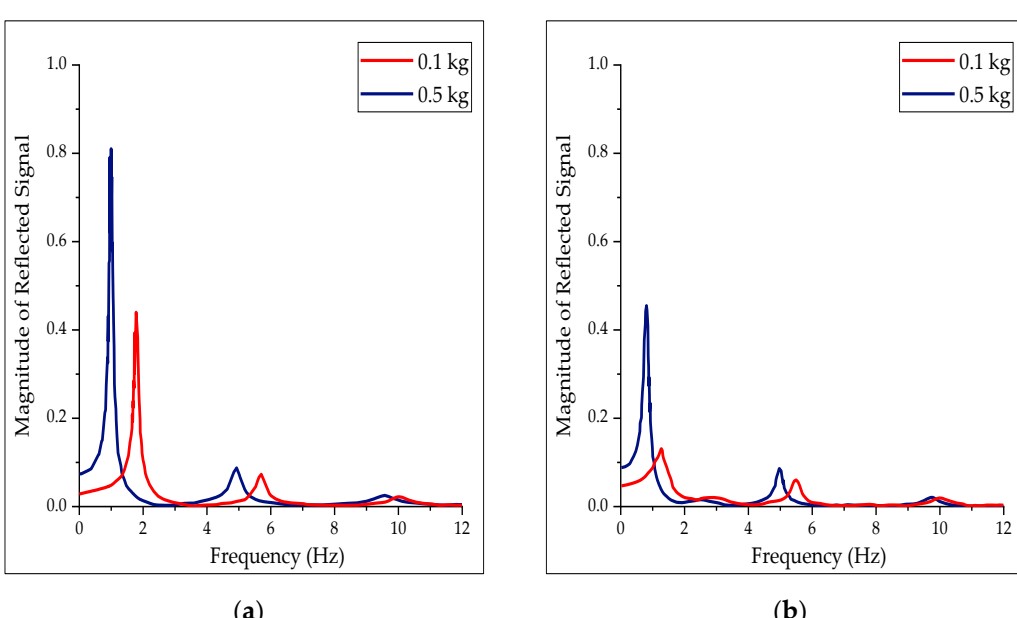

(**a**)　　　　　　　　　　　　　　　　　　　　　　　(**b**)

**Figure 4.** Comparing FFT of the feedback force for masses 0.1 kg and 0.5 kg with 100 ms time delay (Simulations) (**a**) wave variable implemented, and (**b**) shaping wave filter implemented.

Despite the success of the shaping wave filters with respect to time delay, the method struggles when the size of the slave mass is increased from 0.1 kg to 0.5 kg. This is because the shaping wave filters mitigate wave reflections based on the time delay without considering the effect of change in dynamics on the frequency components of wave reflections. Nonetheless, to our knowledge, no previous research has examined this problem.

## 4. Bandstop Wave Filter

In this work, we develop a wave-variable-based controller (bandstop wave filter). In addition, we experimentally verify that the technique can mitigate wave reflections based on the time delay and changes to system dynamics of the teleoperated system. We apply the bandstop wave filter in the wave domain and filtered the wave signal along the communication channel. We placed the bandstop wave filter in the master-to-slave robot path to alleviate lower frequency components of the reflected signal. With the lower frequency components reduced, wave reflections that degrade teleoperation

performance were mitigated and we obtained a better transient response from the system. For a specific range of slave mass, the bandwidth of the filter can be determined. We provide experimental verification of this wave filtering technique using a tele-robot system consisting of two Force Dimension parallel manipulators in Section 5.

From Figure 4a, as the frequency components increase the magnitude of reflected signal reduces. Hence, a better transient response can be obtained from the system by alleviating lower frequency components. As a result, the bandstop wave filter is positioned between the master and slave devices such that the outgoing wave signal, $u_m$, is filtered as shown in Figure 5. The filter can either be placed along with the master-to-slave device path or vice versa because the wave reflections circulate in both directions of the communication channel. The bandstop filter is not placed in the slave-to-master path, because the capability of implementing the bandstop on the master side (local site) is more feasible than the slave side. In the local site, the user can feasibly fine-tune the filter to mitigate wave reflections than to the remote site. The results of placing a filter on both the master-to-slave path and the slave-to-master path did not warrant the more difficult to implement placement on the slave-to-master path.

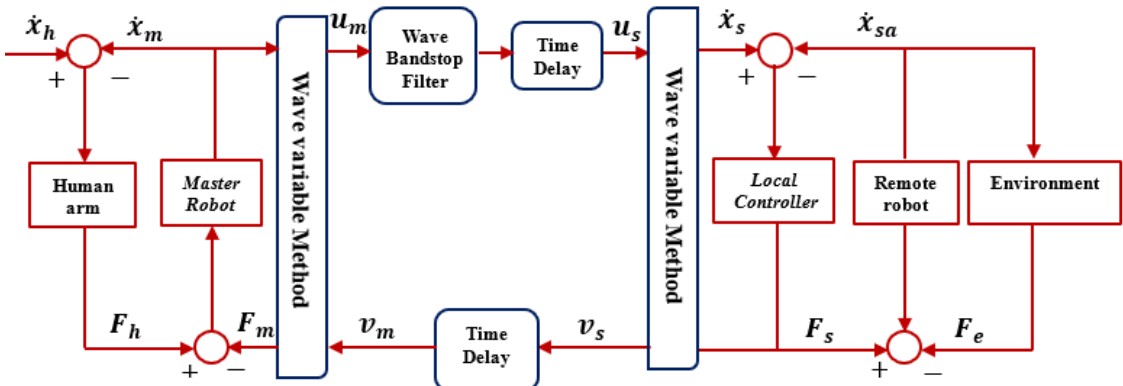

**Figure 5.** Bandstop wave filter and traditional wave variable method implemented in a bilateral teleoperation system.

The bandstop wave filter transfer function can be expressed as [36]:

$$H(s) = \frac{\omega_{c1}}{s + \omega_{c1}} + \frac{s}{s + \omega_{c2}} \tag{21}$$

$\omega_{c1}$ denotes the lower cutoff frequencies and $\omega_{c2}$ denotes the upper cutoff frequencies. Choosing the filter cutoff frequencies depends on the minimum and maximum mass variations anticipated on the slave device. To determine the cutoff frequencies, we tested several percentages of the maximum magnitude of the reflected signal, ranging from 5% to 15%. However, 10% of the maximum magnitude of the reflected signal produced cutoff frequencies that alleviate wave reflections without the loss of important signals. Since the filter operates in the wave domain, Equation (21) can be described as the wave transfer function. It also represents the scattering operator since it illustrates the relationship between wave signals. The implementation of the bandstop filter into the bilateral teleoperation system does not jeopardize its passivity provided the wave transfer function is not greater than 1 [29].

$$\left| H(s) \right| \leq 1 \text{ for all } Re(s) \geq 0 \tag{22}$$

Equation (22) is true provided $\omega_{c1} \leq \omega_{c2}$ from Equation (21). Equation (22) holds for (21) [36]. This implies that the numerator $N(s)$ and denominator $D(s)$ satisfies $\left| D(s) \right|^2 - \left| N(s) \right|^2 \geq 0$ for $Re(s) \geq 0$.

*Simulation Results*

In this section, we compared simulation results from traditional wave variable, shaping wave filter and bandstop wave filter. We set the command velocity of the master device to 0.05 m/s the same value used in Section 3.3. The same teleoperator parameters presented in Table 1 were used.

In Figure 5, we implement the bandstop wave filter in the system illustrated in Figure 3 to see its performance and the simulation results were presented. We applied a second-order Butterworth filter to annihilate the dominant frequencies of the wave reflections. The filter is made up of a high and low cutoff frequency of 2 Hz and 0.5 Hz respectively, giving a corresponding $\omega_{c1}$ and $\omega_{c2}$ of 12.57 rad/s and 3.14 rad/s respectively. These cutoff frequencies were chosen to alleviate wave reflections for slave masses 0.1 kg and 0.5 kg with a time delay of 100 ms.

In Figure 6, we compare the performance of the bandstop wave filter to the shaping wave filtering technique. The dominant frequencies of the wave reflection were mitigated for all the simulated slave masses. From Figure 4a, 0.81 is the highest magnitude when the traditional wave variable was implemented. The highest when the bandstop wave filter was 0.14 as shown in Figure 6b. The magnitude of wave reflection decreased by 83% when the bandstop wave filter was used.

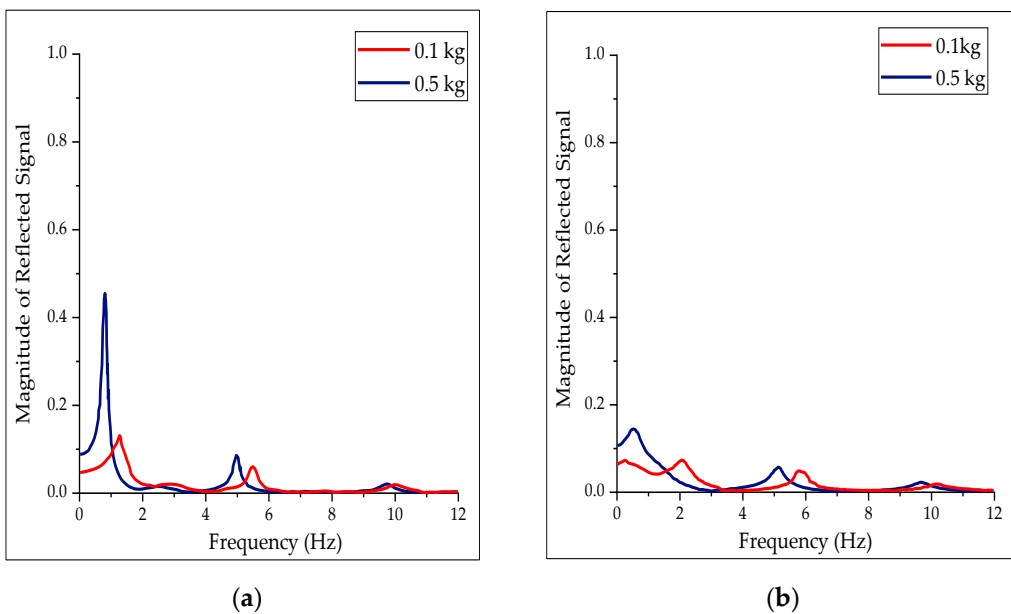

**Figure 6.** Comparing FFT of the feedback force for masses 0.1 kg and 0.5 kg with 100 ms time delay (simulation) (**a**) the shaping wave filter (**b**) bandstop wave filter.

Comparing the performance of the shaping wave filter to the bandstop wave filter, it is apparent that the bandstop wave filter mitigates the dominant frequencies of the wave reflection more than the shaping wave filters. The highest reflected magnitude when shaping wave filter was employed is 0.45 as shown in Figure 6a while 0.14 is the maximum when the bandstop wave filter was used as illustrated in Figure 6b. The magnitude of wave reflection decreased by 69% from 0.45 when the shaping wave filter to 0.14 when the bandstop wave filter was used. Hence, the bandstop wave filter outperforms the shaping wave filter.

To substantiate our simulation results, we compared the performance of the traditional wave variable and shaping wave filter to the bandstop wave filter for 500 ms time delay in Figure 7 and a worst-case simulation result of 5 s time delay in Figure 8.

From Figure 7a, the highest magnitude of the reflected when the traditional wave variable was used is 4.12. The shaping wave filter reduced the highest magnitude of the reflected to 1.44 in Figure 7b. Implementing the bandstop wave filter, we were able to reduce the highest magnitude of the reflected to 0.41 as illustrated in Figure 7c.

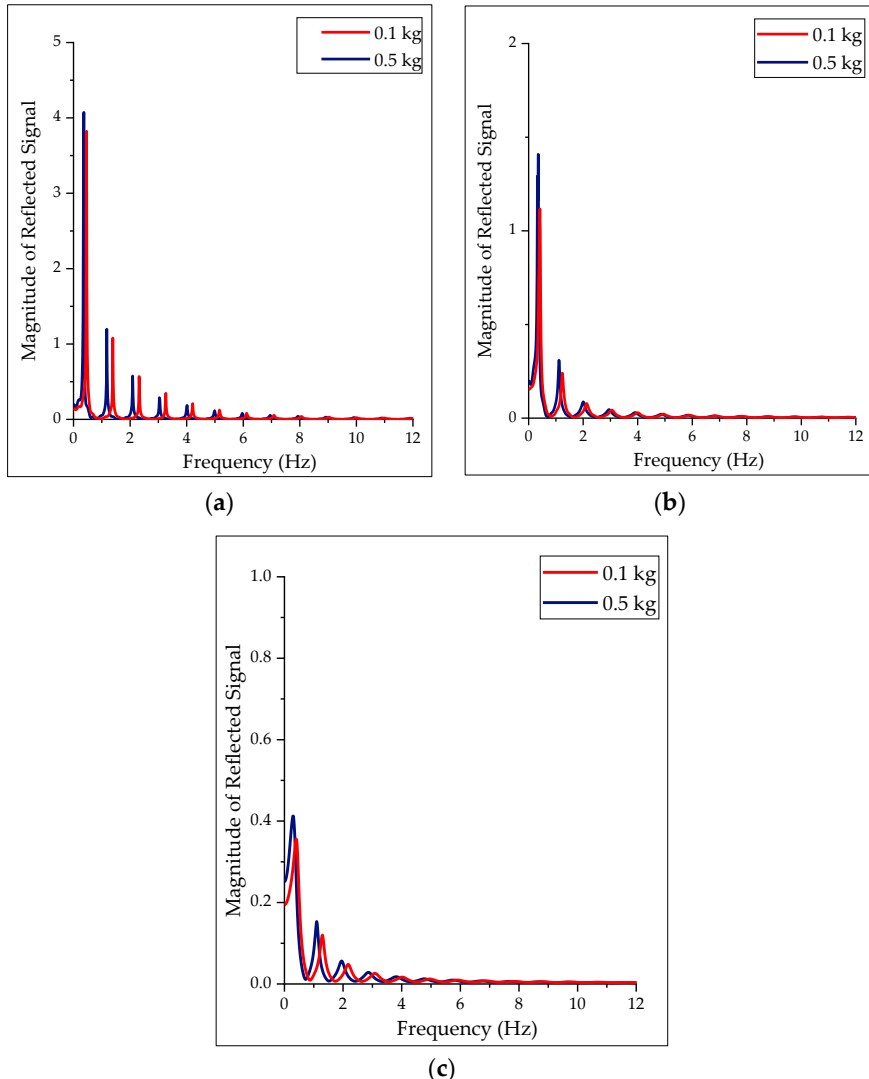

**Figure 7.** Comparing FFT of the feedback force for masses 0.1 kg and 0.5 kg with 500 ms time delay (simulation). (**a**) the wave variable (**b**) the shaping wave filter (**c**) bandstop wave filter.

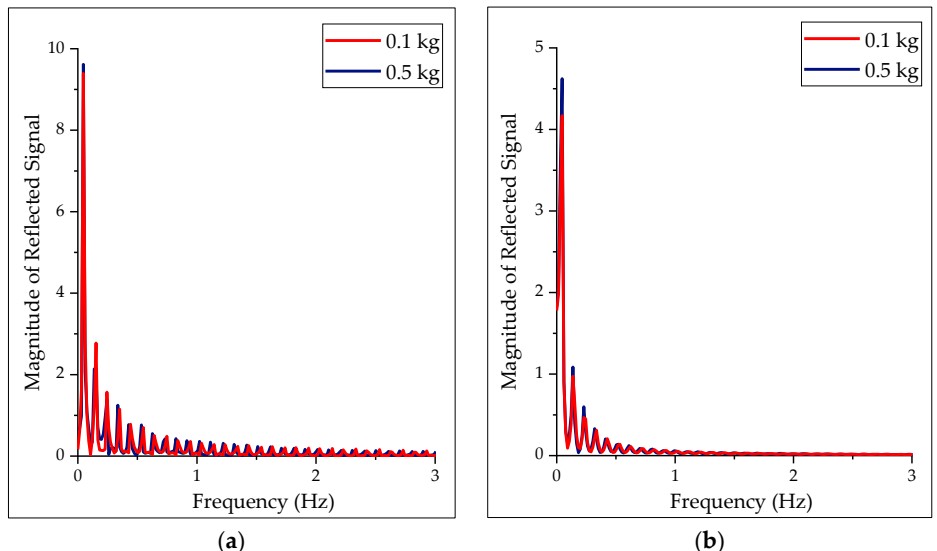

**Figure 8.** *Cont.*

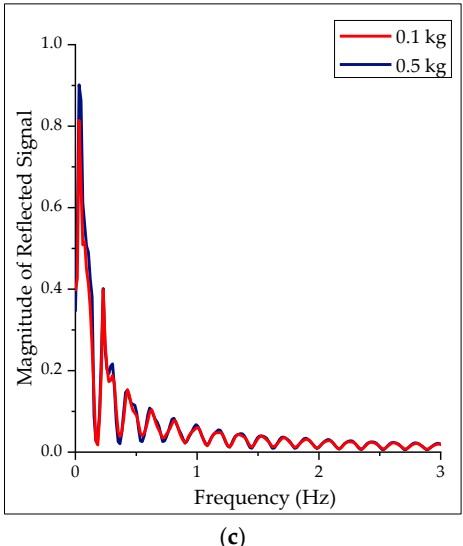

(**c**)

**Figure 8.** Comparing FFT of the feedback force for masses 0.1 kg and 0.5 kg with 5 s time delay (simulation). (**a**) the wave variable (**b**) the shaping wave filter (**c**) bandstop wave filter.

In Figure 8, we provided a worst-case simulation for our proposed bandstop wave filter. 9.6 is the highest magnitude of the reflected signal when the traditional wave variable was used in Figure 8a. With shaping wave filter, the highest magnitude of the reflected signal was reduced to 4.6 in Figure 8b. With the bandstop wave filter implemented, we attenuate the highest magnitude of the reflected signal to 0.9 as illustrated in Figure 8c.

## 5. Experimental Results and Discussion

We provide experimental verification of the wave filtering technique by implementing wave variable-based algorithms on a tele-robot system consisting of two Force Dimension parallel manipulators. Using the position, velocity and force feedback in the experimental results, we compare the bandstop wave filter with the traditional wave variable method and the shaping wave filter.

The master device used in this experiment is a three degree-of-freedom Force Dimension Omega.3 haptic device. This haptic device is a parallel linkage device that provides a closed-loop stiffness of 14.5 N/mm and forces up to 12 N. The slave device used is a Force Dimension sigma.7 with grasping force feedback of ±8 N. It is a seven degree-of-freedom parallel linkage haptic device that has a translational force and torque of 20 N and 0.4 Nm respectively. Both devices interacted via a software development kit called SDK through which the C++ code can be developed. All through the experiment, the slave mass remains fixed during motion and it is being changed only when the slave device is not in motion.

The experimental setup is pictured in Figure 9. Engage the Omega.3 to start the experiment, then the slave device is displaced from its initial position of about −0.05 m from the origin frame to a new position where it is held for a few seconds after which it returns to its origin. On the slave site, a PD controller with proportional and derivative gains of 370 N/m and 2.5 Ns/m respectively is being used. A wave impedance, b, of 2.5 Ns/m is used. We perform this experiment for slave masses 0.1 kg and 0.5 kg. For each of the masses, the time delay of 100 and 500 ms was implemented for the traditional wave variable method, shaping wave filter and bandstop wave filter. Low pass and high pass filter cutoff frequencies for the bandstop wave filter are provided in Table 2. The cutoff frequencies occur at about 10% of the maximum magnitude of the reflected signal. To demonstrate the effectiveness of the bandstop wave filter we compare its performance to that of the traditional wave variable method and the shaping wave filter.

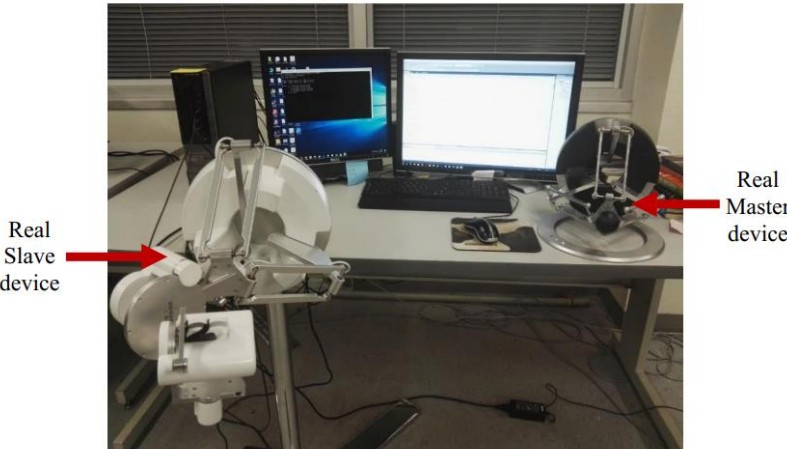

**Figure 9.** Experiment setup showing the Force Dimension Sigma.7 (**Left**) and Omega.3 (**Right**) robots as slave and master devices respectively.

**Table 2.** Lower and upper cutoff frequencies for the bandstop wave filter.

| Slave Mass (kg) | Time Delay (ms) | Cutoff Frequencies (Hz) | |
| --- | --- | --- | --- |
| | | Lower | Upper |
| 0.1 | 100 | 1.28 | 1.69 |
| | 500 | 0.35 | 0.82 |
| 0.5 | 100 | 0.37 | 0.52 |
| | 500 | 0.27 | 0.41 |

To determine settling time when the bandstop wave filter and the shaping wave filter were employed, a step reference input was given to the master device and the response from the slave device was analyzed. The step-response starts at the time when the master device is displaced ($t_{displace}$). The settling time was determined when the response reach and stay within 2% of its final value. For each of the cases, we observed steady-state error in the system because velocity.

Instead of position is the parameter encoded before being transmitted across the communication link. The steady-state response of the slave device does not lie within the 2% of the desired steady-state value as shown in Figure 10. However, to determine the steady-state value of the slave response we choose the tolerance to be 0.0001 which is two orders of magnitude smaller than 2%. The purpose of the tolerance is to create a limit to the deviation of the steady-state. On the slave response, wherever the value of the set gap is less than the tolerance as time goes to infinity is the steady-state value. We calculate the 2% of the determined steady-state value, the corresponding time minus the time at which the slave device was displaced is the settling time. Table 3 illustrates the settling time for the bandstop wave filter and the shaping wave filter. Recall in Section 3 that the shaping wave filters could alleviate wave reflections for limited cases. Therefore, for the slave mass of 0.5 kg at 500 ms, the settling time for the shaping wave filter remains unknown.

**Table 3.** Comparing the settling time for the bandstop wave filter to the shaping wave filter.

| Slave Mass (kg) | Time Delay (ms) | Settling Time (s) | |
| --- | --- | --- | --- |
| | | Shaping Wave Filter | Bandstop Wave Filter |
| 0.1 | 100 | 5.36 | 2.19 |
| | 500 | 17.68 | 12.41 |
| 0.5 | 100 | 26.98 | 3.52 |
| | 500 | - | 17.68 |

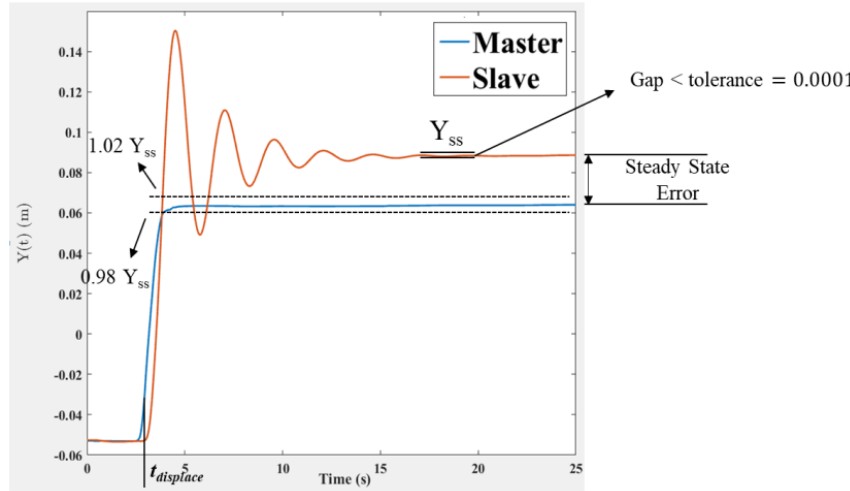

**Figure 10.** Slave device response to a step reference input from the master device.

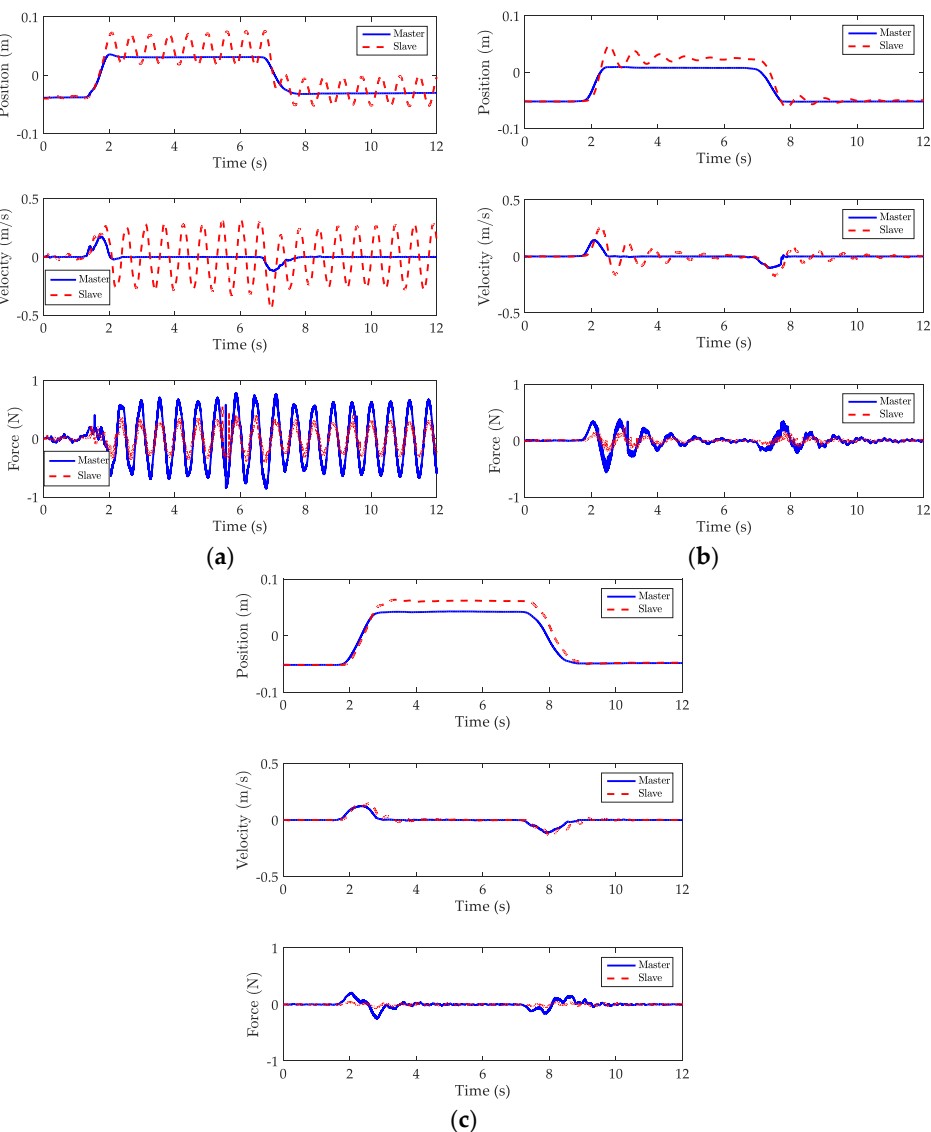

**Figure 11.** Experimental results for the real master and real slave robot of the mass of 0.1 kg at a time delay of 100 ms with the position, velocity and force plotted (**a**) the traditional wave variable (**b**) the shaping wave filter (**c**) the bandstop filter.

Figure 11 illustrates the experimental results when the slave mass is set to 0.1 kg and time delay to 100 ms. The trajectory plots for the traditional wave variable method, shaping wave filter and the bandstop filter are shown in Figure 11a–c respectively. Wave reflections linger when the traditional wave method is implemented (Figure 11a). The performance of the shaping wave filter is comparable with the bandstop wave filter. Recall that the shaping wave filter performs well in comparison to the bandstop wave filter for smaller slave masses because the frequency at which wave reflections occur for smaller mass can be estimated by just the time delay across the communication link. The slave device closely tracks the master device due to smaller mass and time delay in Figure 11b,c. Wave reflections settle at 2.19 s for the bandstop wave filter 5.36 s for the shaping wave filter.

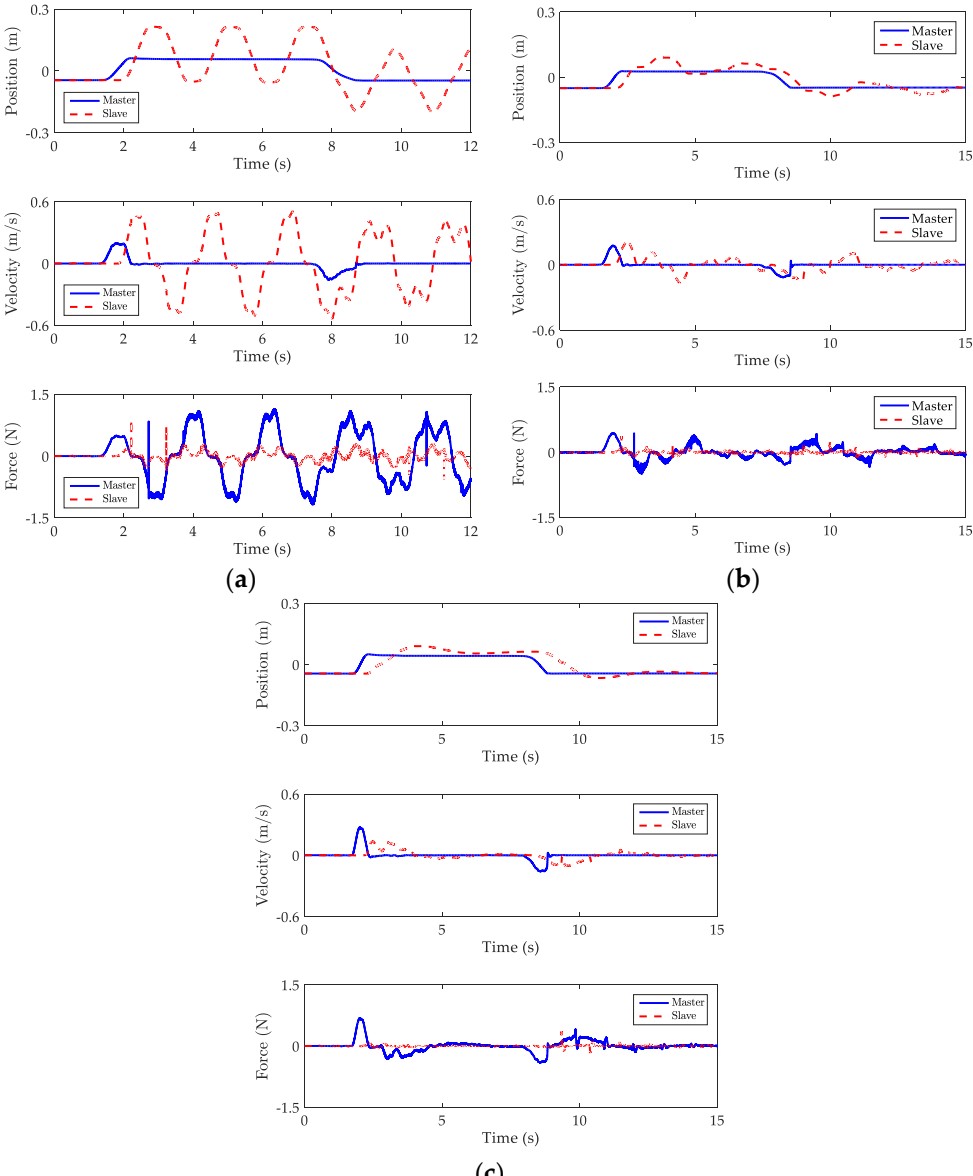

**Figure 12.** Experimental results for the real master and real slave robot of the mass of 0.1 kg at a time delay of 500 ms with the position, velocity and force plotted (**a**) the traditional wave variable (**b**) the shaping wave filter (**c**) the bandstop filter.

We present experimental results when the slave mass and time delay are set to 0.1 kg and 500 ms respectively as shown in Figure 12. The trajectory plots for the traditional wave variable method, shaping wave filter and the bandstop filter are shown in Figure 12a–c respectively. As time delay

increases wave reflection becomes prominent, the period of oscillation grew longer and the gap between successive amplitudes became wider when the traditional wave variable method was implemented in Figure 12a. With the bandstop filter implemented in Figure 12c the wave reflections are attenuated at 12.41 s, while the shaping wave filter attenuated wave reflections at 17.68 s. Comparing the performance of the bandstop wave filter to the shaping wave filter for the slave of 0.1 kg and time 500 ms, evidently, the shaping wave filter took longer to attenuate wave reflections than the bandstop wave filter. Therefore, it can be said that the bandstop wave filter performed better than the shaping wave filter.

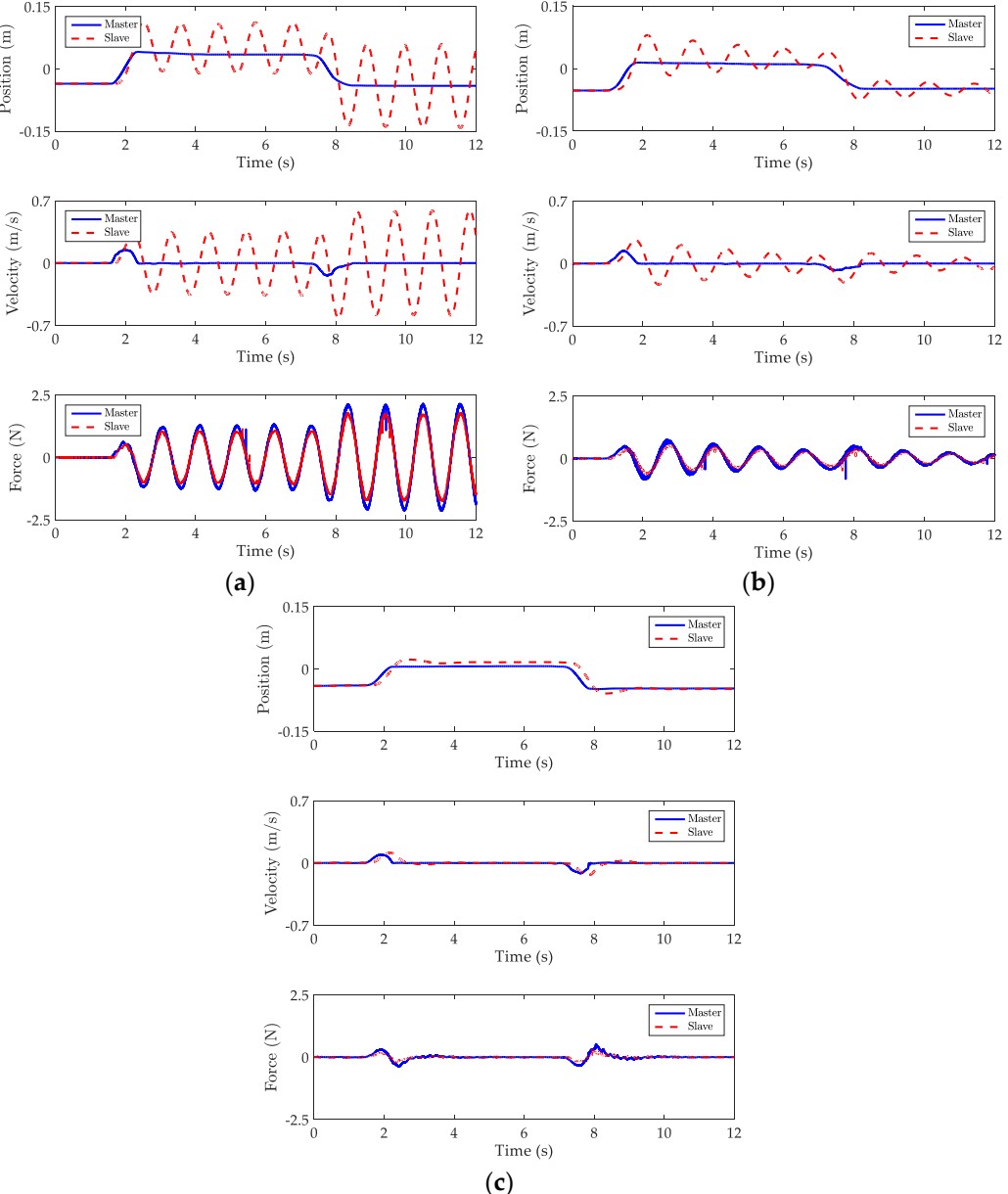

**Figure 13.** Experimental results for the real master and real slave robot of the mass of 0.5 kg at a time delay of 100 ms with the position, velocity and force plotted (**a**) the traditional wave variable (**b**) the shaping wave filter (**c**) the bandstop wave filter.

In Figure 13, we present experimental results for slave mass and time delay of 0.5 kg and 100 ms respectively. The trajectory plots for the traditional wave variable method, shaping wave filter and the bandstop filter are shown in Figure 13a–c respectively. The magnitude of the force feedback grew to about 2 N for a slave mass of 0.5 kg when wave variable was used in Figures 13a and 14a while it

was about 1 N for a slave mass of 0.1 kg in Figures 11a and 12a. This occurs because the larger the mass, the higher the reflected signal and the lower the corresponding frequency components as noted earlier in Section 4. The bandstop wave filter when used in Figure 13c attenuated wave reflections at 3.52 s. We compare the performance of the bandstop filter to the shaping wave filter which attenuates wave reflections at 26.98 s when used in Figure 13b. It is evident that the bandstop filter performs better, while it became difficult for the shaping wave filter to attenuate wave reflection due to the larger slave mass.

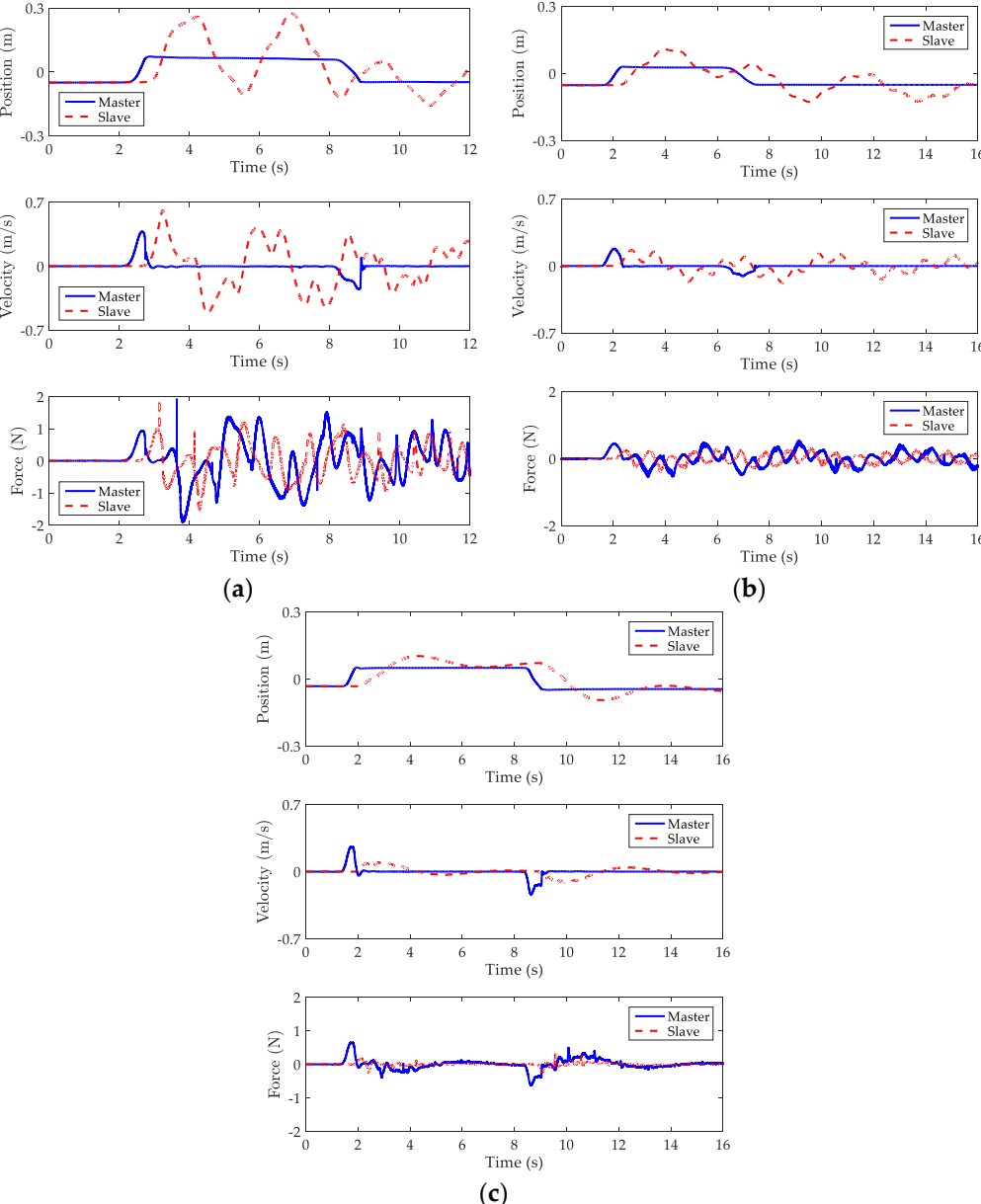

**Figure 14.** Experimental results for the real master and real slave robot of the mass of 0.5 kg at a time delay of 500 ms with the position, velocity and force plotted (**a**) the traditional wave variable (**b**) the shaping wave filter (**c**) the bandstop wave filter.

In Figure 14a,b, the slave device could no longer track the master device due to larger slave mass and increases in time delay. The shaping wave filter struggles seriously for a larger slave mass of 0.5 kg with a time delay of 500 ms, while the bandstop wave filter attenuates wave reflections at 16.5 s. It is apparent that the bandstop wave filters outperformed both the traditional wave variable method and the shaping wave method for a larger slave mass of 0.5 kg with a time delay of 500 ms.

Overall, the bandstop wave filter attenuates wave reflection faster than shaping wave filter for smaller or larger slave mass and time delay.

We present the experimental repeatability of the real master and real slave devices. An experiment is repeatable when the results of subsequent measurements closely agree with the results previously obtained under the same conditions [51,52].

In Figure 15, we present a 95% confidence interval for each of the settling time obtained for the slave mass of 0.1 kg at a time delay of 100 ms and 500 ms. The 95% confidence interval provided a range of settling time within which wave reflections will be attenuated for each of the time delays. For time delay of 100 ms, we are 95% confident that the wave reflections will be mitigated between 2.19 s and 2.29 s when the bandstop wave filter is employed and, 5.27 s and 5.59 s when the shaping wave filter is used.

For a time delay of 500 ms, we are 95% confident that the wave reflections will be mitigated between 12.08 s and 12.84 s when the bandstop wave filter is employed and, 16.75 s and 19.01 s when the shaping wave filter is used. The margin of error increase as time delay increases for both the bandstop wave filter and the shaping wave filter. Also, the margin of error of the shaping wave filter is larger than the bandstop wave filter due to larger settling time. This means that the range of settling time at which wave reflections are attenuated is larger for the shaping wave filter compared to the bandstop wave filter. The bandstop wave filter is precise in attenuating wave reflections than the shaping wave filter.

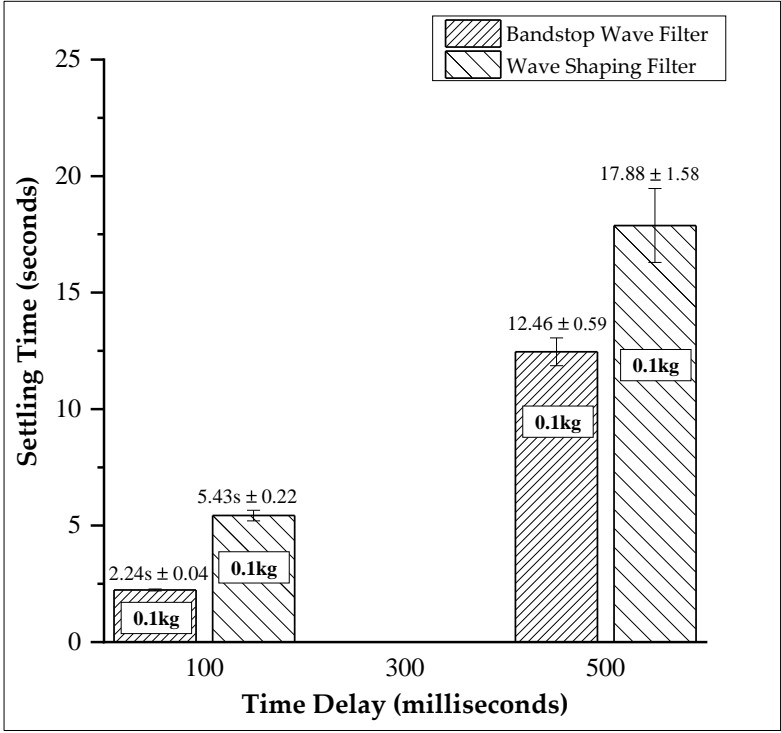

**Figure 15.** Comparing the repeatability of experimental results from the bandstop wave filter with shaping wave filter for a slave mass of 0.1 kg at different time delays.

## 6. Conclusions and Future Work

Here, we presented a bandstop wave filtering technique and experimentally verified its capability to improve teleoperation performance. The contributions of this research may be summarized as follows:

- We developed a bandstop wave filter that can be applied in the wave domain for mitigating reflected wave signals along a communication channel of bilaterally teleoperated systems.

- We were able to alleviate lower frequency components of reflected wave signals by placing the bandstop wave filer in the master-to-slave robot path.
- With the lower frequency components reduced, reflected wave signals (wave reflections) that degrade teleoperation performance was mitigated and we obtained a better transient response from the system.

Although additional experimental testing may be needed to establish the full benefits of the bandstop wave filtering technique for teleoperation systems. Nonetheless, our experimental results provided evidence that the bandstop wave filtering technique may significantly improve teleoperation performance in applications including robotic surgery, search and rescue robots, teleoperated military combat drones, industrial robots, space exploration, etc.

The cut-off frequencies of the bandstop wave filter implemented in this work need to be tuned by the operator every time the range of slave mass changed. This is needful to target the most dominant frequencies of the wave reflections for the new range of slave mass. However, in the future, a self-tuning filter or adaptive filter that can mitigate wave reflections when the slave mass is change can be developed. To achieve this, the frequency of wave reflections can be determined based on the slave mass and implemented in the adaptive filter as a reference. Based on this reference, the adaptive filter will be able to tune itself to mitigate wave reflections.

**Author Contributions:** C.F.A. developed the theory and performed simulations. I.O.O., C.A.M.J., R.G.R. and K.M. planned the experiments. K.M. and I.O.O. wrote the C++ program used for the experiments. I.O.O. performed the experiments. C.A.M.J. and R.G.R. helped to supervise the project. I.O.O. wrote the manuscript with contributions from C.F.A., C.A.M.J. and R.G.R. All authors have read and agree to the published version of the manuscript.

**Funding:** This research received no external funding.

**Conflicts of Interest:** The authors declare no conflict of interest.

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
