# Peer review of "Experimental Testing of Bandstop Wave Filter to Mitigate Wave Reflections in Bilateral Teleoperation"

_robotics, doi:10.3390/robotics9020024_

Round 1

Reviewer 1 Report

This paper introduces a filtering approach to deal with the wave reflection problem. My comments are as follows

  1. In (18), the modeled human force is not fit for reality. First, it should contain acceleration information. Second the gians K_h and B_h are usually variable and unknown.
  2. The authors manually set a high and low cutoff frequency of 2Hz and 0.5Hz respectively, giving a corresponding ??1 and ??2 of 12.57 rad/sec and 3.14 rad/sec respectively. Why? The authors should mathematically prove a range that can offer the system best performance. Otherwise, simply setting a number makes the contribution too vague. Afterall the proposed filter is just a first order low-pass filter plus a first-order high-pass filter.
  3. Since the wave variable method is used to deal with large time delays. The authors should provide more experiment results on different delays. Especially, the delays in worst case (>2s) should be considered.
  4. The references are not enough. Please analysis the following papers on wave reflections

1. Application of wave-variable control to bilateral teleoperation systems: A survey. Annual Reviews in Control, 38(1), 12-31.

2. Reducing wave-based teleoperator reflections for unknown environments. IEEE Transactions on Industrial Electronics, 58(2), 392-397.

3. A novel approach for stability and transparency control of nonlinear bilateral teleoperation system with time delays. Control Engineering Practice, 47, 15-27.

4. Improving haptic feedback fidelity in wave-variable-based teleoperation orientated to telemedical applications. IEEE Transactions on Instrumentation and Measurement, 58(8), 2847–2855.

5. Bilateral teleoperation with reducing wave-based reflections. Advances in Manufacturing, 1(3), 288–292.

Author Response

Dear Reviewer,

Reviewer 2 Report

The paper presents a Teleoperation System stabilized by scattering transformation. The novelty of the paper is the use of a cutoff filter to improve the settling time of the step response by filtering the wave reflections due the wave variables.
The idea is good and the topic is of interest. Nevertheless, in my opinion it is more suitable for a conference paper than for a journal paper, mainly due the method simplicity and several required improvements.

Some remarks:

  • why is the bandstop filter not placed also in the Slave2Master path? Effects if placing the Filter in the Slave2Master channel in addition to the Master2Slave path.
  • It does not solves the variable delay problem ...
  • In my opinion, the proof of passivity for wave variables are not necessary. It is already presented in several previous studies and very well known for constant delay in the communication channel.
  •  I recommend to use the nomenclature "u_x, w_x" in place of "u_x, v_x" for the wave variables because of possible confusion with velocity. It is only a suggestion.
  • It is a good idea, but in my opinion the paper is more suitable as a conference paper than for a journal paper.
  • Please explain how to select the cutoff frequencies w_{c1} and w_{c2}; if you follow a method or it is an heuristic result.
  • Comparison are made between the Cutoff Wave Filter and a Shaping Wave Filter, but this one is not presented, neither its parameters and the relation with the communication delay.
  • Figure 12 is bad numbered!! ... In figure 12, the step-response is started at which time? ... are all the settling times referenced to the moment when the input signal steeps-up?

- Typo:
L314 "um" should be in math form.
L347 "equation (20) is holds for (19)" ... Please explain this sentence.

Author Response

Dear Reviewer,

Round 2

Reviewer 1 Report

I have no more comments.